# Language as a Wave Phenomenon: Semantic Phase Locking and Interference in Neural Networks

**Alper Yıldırım** [1]  **İbrahim Yücedağ** [2]

## Abstract

In standard Transformer architectures, semantic importance is often conflated with activation magnitude, obscuring the geometric structure of latent representations. To disentangle these factors, we introduce PRISM, a complex-valued architecture designed to isolate the computational role of phase. By enforcing a strict unit-norm constraint ($|z| = 1$) and replacing attention with gated harmonic convolutions, the model is encouraged to utilize subtractive interference in the frequency domain to suppress noise, rather than relying on magnitude-based gating. We utilize this constrained regime to study a hybrid architecture—fusing phase-based routing with standard attention—which achieves improved parameter efficiency and representation quality compared to baselines in our evaluated settings. Mechanistically, interventional ablations indicate that the model carries substantial task-relevant information in phase: preserving phase largely maintains performance, whereas disrupting phase causes severe degradation. Together, these results suggest that phase-based spectral interference is a usable computational mechanism for neural sequence modeling at the evaluated scale.

## 1. Introduction

A central question in neural sequence modeling is which computational primitives are necessary rather than conventional. In standard real-valued architectures such as Transformers (Vaswani et al., 2017), semantic importance is often coupled to activation magnitude: signals are selected, suppressed, or emphasized largely through changes in scale. Complex-valued networks (Trabelsi et al., 2018) offer a different primitive. They allow information to be encoded not only in magnitude, but also in phase, enabling interactions through rotation and interference rather than purely additive accumulation.

However, in unconstrained complex-valued networks, phase and magnitude remain entangled. A model may still encode salience by amplifying energy, making it difficult to determine whether phase is an active computational carrier or merely a representational byproduct. We therefore study a constrained spectral setting in which global mixing occurs through complex filtering and phase rotation, while magnitude-based shortcuts are deliberately limited. Our goal is not to claim that language is literally a wave phenomenon, but to use wave mechanics as a computational lens for studying complex-valued sequence models.

We introduce the Phase-Rotating Interference Spectral Model (PRISM), a complex-valued architecture inspired by optical 4f-correlators. PRISM replaces attention in its spectral branch with an FFT$\rightarrow H \rightarrow$IFFT filtering pipeline, allowing learned complex filters to shape sequence representations through phase-sensitive interference. Unlike real-valued spectral mixers such as FNet, which perform additive Fourier mixing, PRISM maintains persistent complex representations and learns phase rotations across layers.

Our experiments ask whether this phase structure is functionally useful. Our experiments proceed in two stages. First, we use WMT14 as a hypothesis-guided probing setting to examine whether phase coherence and phase rotation exhibit the qualitative structure predicted by phase-based semantic coding. These experiments are not intended as the main benchmark evidence; rather, they motivate the causal tests that follow. Second, we evaluate PRISM-style encoder models on WikiText-103 and perform interventional ablations on the learned spectral pathway. These ablations show that preserving spectral-filter phase retains most performance, whereas removing, reversing, or shuffling phase causes severe degradation.

**Scope & Scale.** This work investigates the role of phase in neural sequence modeling. We use controlled architectures and interventional ablations to test whether phase carries

[1]Independent Researcher, Türkiye [2]Department of Computer Engineering, Düzce University, Düzce, Türkiye. Correspondence to: Alper Yıldırım <yildirim.alper.dev@gmail.com>, İbrahim Yücedağ <ibrahimyucedag@duzce.edu.tr>.

*Proceedings of the $43^{rd}$ International Conference on Machine Learning*, Seoul, South Korea. PMLR 306, 2026. Copyright 2026 by the author(s).

task-relevant information in learned sequence representations.

## 2. Related Work

**Spectral & Harmonic Modeling.** FNet (Lee-Thorp et al., 2022) established that Fourier Transforms could replace attention with $\mathcal{O}(N \log N)$ efficiency, though it relies on additive magnitude mixing. Recent work in continuous physics, such as CoNO (Tiwari et al., 2025), demonstrated that complex-valued operators are essential for capturing rotational dynamics. PRISM is related to this line of work, but applies complex-valued spectral operations to discrete sequence modeling.

**Phase-Based Attention.** While Holographic Transformers (Huang et al., 2025) and POC-ViT (Sharma et al., 2025) utilize phase to achieve signal robustness and invariance, they retain the quadratic complexity of attention. Complex Vector Attention (Shao et al., 2025) further demonstrates the utility of complex-valued representations within attention mechanisms. In contrast, PRISM replaces attention entirely, utilizing phase for *semantic steering*—participating in ambiguity resolution via subtractive interference—within a globally efficient spectral framework.

**Unitary Evolution & Hardware.** The theoretical basis for our iso-energetic constraint lies in Unitary RNNs (Arjovsky et al., 2016), where Arjovsky et al. proved that unity gain enables infinite memory retention. We adapt this to encode information in phase angles, effectively simulating the Phase-Coded Aperture hardware topology established by Chi & George (Chi & George, 2011). Finally, we reject covariance-based Complex Batch Normalization (Trabelsi et al., 2018) in favor of Phase-Preserving Layer Normalization (PPLN) to prevent phase distortion during optimization.

## 3. Architecture

We formally define the Phase-Rotating Interference Spectral Model (PRISM), a structural modeling paradigm rooted in wave mechanics. Unlike standard Transformers which treat tokens as static vectors manipulated by additive updates, PRISM treats tokens as complex phasors $z = re^{i\theta}$, where semantic identity is represented primarily through the angle $\theta$ and signal intensity through the magnitude $r$.

### 3.1. Rotary Semantic Embeddings (RoSE)

We introduce Rotary Semantic Embeddings (RoSE) to unify semantic identity and positional context into a single complex-valued phasor. Unlike standard architectures that add positional encodings to static embeddings ($e + p$), RoSE operates via multiplicative rotation in the complex plane.

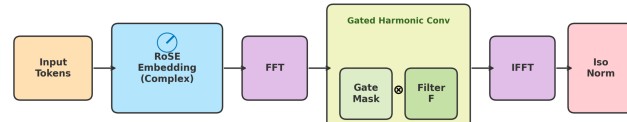

*Figure 1.* The PRISM Architecture. (Left) Input tokens are encoded as complex phasors via RoSE. (Center) Gated Harmonic Convolutions replace $\mathcal{O}(N^2)$ attention with $\mathcal{O}(N \log N)$ spectral filters $\mathbf{H}$. (Right) Iso-Energetic normalization encourages phase-based encoding within the spectral branch.

Let $w_t$ be the token at position $t$. We first project $w_t$ into a complex latent space $\mathbb{C}^d$ by learning separate real and imaginary components, forming a content vector $z_t \in \mathbb{C}^d$. To encode position, we define a spectrum of geometric frequencies $\boldsymbol{\omega} \in \mathbb{R}^d$, distributed logarithmically to capture multi-scale dependencies:

$$\omega_k = \frac{1}{10000^{k/d}}, \quad k \in [0, d-1] \quad (1)$$

The final embedding $E(w_t)$ is obtained by rotating the content vector $z_t$ by the positional phase angle:

$$E(w_t) = z_t \cdot e^{i\boldsymbol{\omega}t} \quad (2)$$

This formulation ensures that the relative distance between tokens $t$ and $t+k$ is preserved as a constant phase shift $e^{i\boldsymbol{\omega}k}$ in the frequency domain, providing a natural inductive bias for sequence modeling.

### 3.2. Gated Harmonic Convolution

The core reasoning unit of PRISM is the Gated Harmonic Convolution. Unlike recent Holographic architectures which introduce phase dynamics but retain the quadratic computational cost of pairwise attention (Huang et al., 2025), PRISM replaces the attention mechanism entirely. This layer filters information in the frequency domain ($\mathcal{O}(N \log N)$) while allowing the time-domain signal to dynamically modulate its own phase and magnitude.

Given a sequence of complex embeddings $\mathbf{X} \in \mathbb{C}^{L \times d}$, we first normalize the input using Phase-Preserving Layer Normalization. We then compute two separate data-dependent gates, $\mathbf{G}_{re} \in \mathbb{R}^{L \times d}$ and $\mathbf{G}_{im} \in \mathbb{R}^{L \times d}$, using the concatenated real and imaginary components of the normalized input $\tilde{\mathbf{X}}$:

$$[\mathbf{G}_{re} \| \mathbf{G}_{im}] = \sigma\left(\mathbf{W}_{gate}[\text{Re}(\tilde{\mathbf{X}}) \| \text{Im}(\tilde{\mathbf{X}})]\right) \quad (3)$$

Simultaneously, we transform the sequence into the frequency domain via the Fast Fourier Transform (FFT) and apply a learnable global filter $\mathbf{H} \in \mathbb{C}^{L \times d}$. This operation provides global sequence mixing through a complex-valued spectral filter, allowing each frequency component to be modulated before returning to the time domain. It is related to prior efficient spectral and state-space sequence models (Lee-Thorp et al., 2022; Gu & Dao, 2024), but differs by

maintaining persistent complex-valued representations and learned phase rotations across layers.

$$\mathbf{Y}_{freq} = \text{FFT}(\tilde{\mathbf{X}}) \odot \mathbf{H} \qquad (4)$$

The filtered signal is returned to the time domain via the Inverse FFT (IFFT). Crucially, we apply **Cartesian Gating**, where the real and imaginary components are modulated independently:

$$\begin{aligned} \mathbf{Y}_{out} = \text{Re}(\text{IFFT}(\mathbf{Y}_{freq})) \odot \mathbf{G}_{re} \\ + i \cdot (\text{Im}(\text{IFFT}(\mathbf{Y}_{freq})) \odot \mathbf{G}_{im}) \end{aligned} \qquad (5)$$

Unlike scalar gating, which only scales magnitude, this independent Cartesian modulation allows the network to alter the phase angle $\theta = \arctan(\text{Im}/\text{Re})$ of the latent vectors, enabling content-dependent phase modulation.

### 3.3. Phase-Preserving Non-Linearity (ModReLU)

Standard activation functions like ReLU are ill-defined for complex numbers as they destroy phase information. We adopt ModReLU, which rectifies the magnitude of the complex vector while strictly preserving its phase angle. For a complex input $z$:

$$\text{ModReLU}(z) = \text{ReLU}(|z| + b) \cdot \frac{z}{|z|} \qquad (6)$$

where $b$ is a learnable bias parameter. This ensures that the semantic orientation of the vector (its "meaning") remains unchanged, while its intensity (magnitude) is non-linearly scaled.

### 3.4. Phase-Preserving Dropout

To prevent the network from over-relying on specific frequency bands, we introduce Phase-Preserving Dropout. Unlike standard dropout which operates on scalars, applying independent masks to the real and imaginary components would destroy the phase angle $\theta$. Instead, we sample a single binary mask $m \sim \text{Bernoulli}(1 - p)$ and apply it identically to both components:

$$z_{drop} = (m \cdot \text{Re}(z)) + i \cdot (m \cdot \text{Im}(z)) \qquad (7)$$

This ensures that if a feature is dropped, its entire spectral contribution (amplitude and phase) is removed simultaneously, forcing the network to distribute semantic information holographically across the spectrum.

### 3.5. Phase-Preserving Layer Normalization

We normalize complex activations by applying LayerNorm to their magnitudes while retaining the original complex direction:

$$\text{PPLN}(z) = \frac{z}{|z| + \epsilon} \odot \left( \frac{|z| - \mu}{\sqrt{\sigma^2 + \epsilon}} \odot \gamma + \beta \right). \qquad (8)$$

This separates magnitude normalization from phase direction. Since LayerNorm can produce negative normalized magnitudes, this WMT14 variant may introduce $\pi$ phase flips. For the WikiText-103 encoder experiments, we use an RMS-based variant that rescales by a positive normalization factor and avoids these sign flips.

### 3.6. Complex-to-Real Bridge

To interface the complex encoder with the real-valued decoder, we utilize a **Projection Bridge**. The state $z_{enc}$ is mapped to $\mathbb{R}^d$ by concatenating its real and imaginary components and projecting via a linear layer $W_{bridge}$:

$$h_{real} = \text{LN}(W_{bridge}[Re(z_{enc})||Im(z_{enc})]) \qquad (9)$$

This compresses spectral phase information into a dense representation suitable for cross-attention.

## 4. Methodology

### 4.1. Dataset and Preprocessing

We evaluate on the WMT14 De-En benchmark (Bojar et al., 2014; Vaswani et al., 2017), tokenized using the Helsinki-NLP OPUS-MT model (Tiedemann & Thottingal, 2020). To reduce computational overhead, we filter sequences exceeding $L = 128$, discarding $< 1\%$ of the data. Training utilizes dynamic bucketing (width 4, target size 20,000) to minimize padding and stabilize spectral gradients.

### 4.2. Experimental Design: Phase-Coding vs. Rate-Coding Regimes

We utilize WMT14 as an initial mechanistic debugging workbench; the primary encoder evaluation and interventional ablations are conducted on WikiText-103 in Section 6. Rather than maximizing metrics via scale, this experiment employs the **most restricted version** of PRISM to verify if semantic relationships (e.g., synonyms) naturally emerge as phase shifts within a controlled environment.

We establish a comparative study between two regimes. The **Control Group (Rate-Coding)** is a standard Transformer (Vaswani et al., 2017) utilizing Rotary Position Embeddings (RoPE) and unconstrained magnitude. The **Experimental Group (Phase-Coding)** is the restricted PRISM architecture, constrained by Iso-Energetic Unity Gain. Both models use 6-layer encoders/decoders and Pre-Layer Normalization (Pre-LN) (Nguyen & Salazar, 2019) for stability.

**Spectral Control (FNet):** We select FNet (Lee-Thorp et al., 2022) as the mathematically isomorphic control. Both rely

on global Fourier mixing, though we restrict PRISM to 1D temporal interference (unlike FNet's 2D mixing) to strictly isolate phase-based reasoning. We exclude causal SSMs (Gu & Dao, 2024) to avoid confounding recurrence variables.

**Fairness & Capacity:** To accommodate FNet's fixed-length training requirement ($L = 128$), we synchronized effective token counts across models. Furthermore, since FNet lacks learnable mixing weights, we matched parameter counts by increasing its encoder depth to 7 layers (vs. 6 for PRISM). This ensures the baseline possesses superior algorithmic capacity (14.7M vs 13.0M), making it unlikely that PRISM's efficiency is due simply to under-parameterizing the control.

**Parameter Redistribution:** PRISM redistributes density from "logic" to "memory." The PRISM Encoder requires **31.2% fewer parameters** than the Standard Transformer (13.0M vs 18.9M). We evaluate the **PRISM-Standard (Tied)** configuration to demonstrate this efficiency is achievable without compromising fidelity.

*Table 1.* **Architectural Configurations.** PRISM (Tied) reduces encoder density by $> 10\%$ compared to FNet.

| COMPONENT | TRANSF. | FNET | PRISM (T) | PRISM (U) |
|---|---|---|---|---|
| EMBEDDINGS | 29.7M | 30.3M | 30.3M | **89.2M** |
| ENCODER | 18.9M | 14.7M | **13.0M** | 13.4M |
| BRIDGE | 0.0M | 0.5M | 0.5M | 0.5M |
| DECODER | 25.2M | 25.2M | 25.2M | 25.2M |
| TOTAL | **73.8M** | **71.2M** | **69.1M** | **128.4M** |

*Debugging Prototype:* We utilize a Sequence-to-Sequence framework with a standard Transformer decoder. This isolates the experimental Phase Coding variable strictly to the reasoning encoder before scaling to the high-capacity architectures in Section 6.3.

### 4.3. Training Configuration

All models were optimized using AdamW (Loshchilov & Hutter, 2019) with weight decay of 0.01 and a cosine learning rate schedule (600 warmup steps).

**Strict Hyperparameter Parity:** To ensure a rigorous evaluation, we enforced **identical hyperparameters** across all comparative benchmark models (Transformer, FNet, and PRISM-Tied).

**Stability Constraints:** Initial experiments revealed that while the PRISM architecture remained stable at higher learning rates ($8 \cdot 10^{-4}$), the Standard Transformer control group exhibited gradient instability. Consequently, we restricted the peak learning rate to $6 \cdot 10^{-4}$ for all models, effectively handicapping PRISM to accommodate the stability limits of the Transformer baseline.

**Spectral Precision:** Global batch size was maintained at approximately 20,000 tokens via gradient accumulation. We utilized Single Precision (FP32) to prevent the catas-

trophic cancellation of phase information inherent in Mixed-Precision (FP16) arithmetic.

### 4.4. Hypotheses

We test three predictions that follow from the premise that phase angles encode semantic information under the unit-norm constraint:

**Phase coherence reflects semantic relationships.** If phase encodes semantic content, then word pairs with known semantic relationships (synonyms, antonyms) should exhibit higher phase coherence than unrelated word pairs. We measure this via the Weighted Mean Resultant Length $R$ (Section 4.5).

**Ambiguity resolution occurs through phase rotation.** A polysemous token (e.g., "bank") admits multiple valid semantic interpretations, each corresponding to a different region in phase space. If phase participates in this ambiguity-resolution process, then polysemous tokens should undergo larger phase rotations than unambiguous tokens at specific layers. We test this on a contrastive set of ambiguous and unambiguous tokens.

**Phase-based computation requires multi-token interference.** A single token produces only a DC component under FFT; a token pair produces only two frequency bins. In both cases, the spectral resolution is insufficient for the harmonic filters to operate. We predict that the model will fail to produce coherent output when sequence length falls below a minimum threshold.

### 4.5. Measurements

We instrument each encoder layer $l$ with forward hooks, capturing the input $x^{(l)}$ and output $y^{(l)}$. We define the following metrics:

**Signal Gain ($g$).** The ratio of output to input magnitude, measuring whether a layer amplifies or preserves signal energy:

$$g^{(l)} = \frac{\|y^{(l)}\|_2}{\|x^{(l)}\|_2 + \epsilon} \qquad (10)$$

A value of $g \approx 1.0$ indicates the layer preserves magnitude; $g \gg 1.0$ indicates amplification.

**Phase Rotation ($\Delta\theta$).** The angular displacement between input and output phasors, measuring how much a layer changes the orientation of the representation in the complex plane:

$$\Delta\theta^{(l)} = \arccos\left(\frac{\text{Re}(\langle x^{(l)}, y^{(l)}\rangle)}{\|x^{(l)}\|_2 \|y^{(l)}\|_2}\right) \cdot \frac{180}{\pi} \qquad (11)$$

A large $\Delta\theta$ indicates the layer substantially reoriented the token; a small $\Delta\theta$ indicates the token passed through largely unchanged.

**Rotation Skewness ($\gamma_1$).** The skewness of the phase rotation distribution across tokens, measuring whether the layer applies uniform transformations or selectively rotates a subset of tokens:

$$\gamma_1 = \frac{\frac{1}{n}\sum_{i=1}^{n}(\Delta\theta_i - \bar{\theta})^3}{\left(\frac{1}{n}\sum_{i=1}^{n}(\Delta\theta_i - \bar{\theta})^2\right)^{3/2}} \qquad (12)$$

Near-zero skewness ($\gamma_1 \approx 0$) indicates a uniform transformation across all tokens. High positive skewness ($\gamma_1 > 1.0$) indicates that the layer applies large rotations to a small subset of tokens while leaving the majority unchanged.

**Phase Coherence ($R$).** In real-valued embeddings, cosine similarity measures the angular alignment between two vectors and has served as the standard probe of semantic structure since word2vec. We generalize this to the complex plane: for a pair of token embeddings $z_a, z_b \in \mathbb{C}^d$, we measure the Weighted Mean Resultant Length of their phase differences across frequency bands:

$$R = \left| \frac{\sum_{k=1}^{d} |z_{a,k}||z_{b,k}|e^{i\Delta\phi_k}}{\sum_{k=1}^{d} |z_{a,k}||z_{b,k}|} \right| \qquad (13)$$

where $\Delta\phi_k = \text{Angle}(z_{a,k}) - \text{Angle}(z_{b,k})$. Where cosine similarity operates on a single angle between real-valued vectors, $R$ aggregates phase differences across $d$ frequency bands. $R \in [0, 1]$ quantifies angular alignment: $R = 1$ indicates perfect phase locking across all frequency bands, $R = 0$ indicates uniform random phase differences. The magnitude weighting $|z_{a,k}||z_{b,k}|$ ensures that coherence is measured primarily in high-energy spectral bands.

## 5. Phase Analysis on WMT14

We evaluate the Static RoSE configuration of PRISM on WMT14 De-En as a controlled probe of phase structure. This configuration uses fixed positional phase rotations without content-dependent steering, isolating the baseline behavior of spectral interference. It achieves 0.799 COMET, trailing FNet (0.805) and the Transformer ceiling (0.821). Under the Pre-LN topology used by all models, signal gain remains within 1.0–1.05 across all layers, as expected given the normalize-before-residual design.

### 5.1. Phase Coherence Reflects Semantic Relationships

We constructed a test set of single-token word pairs across three categories: synonyms (e.g., *schnell–rasch*), antonyms (e.g., *Sommer–Winter*), and random pairs (e.g., *Haus–Tisch*). After deduplication, $N = 107$ pairs remained. All words

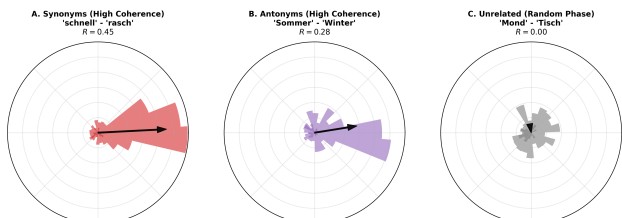

*Figure 2.* Distribution of phase differences $\Delta\phi$ for three representative pairs. **(A)** Synonyms exhibit concentrated phase differences, indicating high coherence. **(B)** Antonyms also show non-random phase structure. **(C)** Unrelated words display a uniform distribution.

*Table 2.* Phase coherence statistics ($N = 107$, deduplicated). Bootstrap 95% confidence intervals are non-overlapping across all categories. All pairwise differences are significant (Mann-Whitney $U$ with Bonferroni correction; permutation test $p < 10^{-4}$).

| CATEGORY | MEAN $R \pm$ STD | 95% CI |
|---|---|---|
| SYNONYM | $0.197 \pm 0.096$ | $[0.172, 0.227]$ |
| ANTONYM | $0.133 \pm 0.064$ | $[0.114, 0.155]$ |
| RANDOM | $0.071 \pm 0.029$ | $[0.059, 0.082]$ |

were verified to produce single tokens under the model's tokenizer. We computed the phase coherence $R$ (Eq. 13) for each pair from the trained embeddings.

As shown in Table 2 and Figure 2, synonym pairs exhibit the highest phase coherence ($R = 0.197$), followed by antonyms ($R = 0.133$), with random pairs forming a distinct noise floor ($R = 0.071$). All pairwise differences are significant after Bonferroni correction (Synonym vs. Random $p < 10^{-7}$; Antonym vs. Random $p = 2.2 \times 10^{-4}$).

Notably, antonyms show higher coherence than random pairs rather than lower. This suggests that phase coherence reflects *semantic relatedness* rather than similarity—antonyms share a topical domain (e.g., temperature for *hot–cold*) even though their meanings oppose.

### 5.2. Ambiguity Resolution via Phase Rotation

We constructed a contrastive dataset of $N = 146$ single tokens divided into polysemous tokens (words with multiple distinct meanings) and unambiguous tokens. We measured phase rotation $\Delta\theta^{(l)}$ at each encoder layer for both groups.

Figure 3 shows that polysemous tokens undergo substantially larger phase rotations than unambiguous tokens, with the divergence concentrated at Layer 3. The rotation distribution for polysemous tokens is heavy-tailed ($\gamma_1 = 1.59$), indicating that the layer applies large corrections to a subset of tokens while leaving others relatively unchanged. Unambiguous tokens show a more symmetric distribution ($\gamma_1 = 0.91$).

We additionally verified that uniformly attenuating spectral

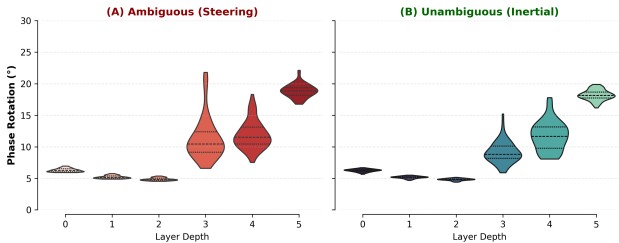

*Figure 3.* Phase rotation by layer. Polysemous tokens (red) show a sharp increase in rotation skewness ($\gamma_1 = 1.59$) at Layer 3, compared to unambiguous tokens ($\gamma_1 = 0.91$).

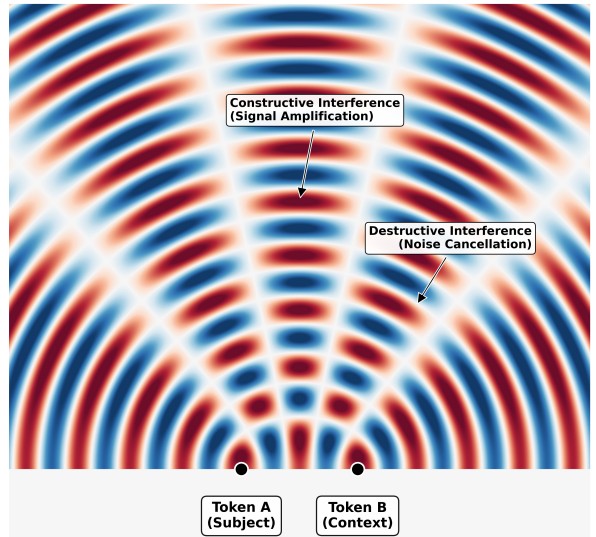

*Figure 4.* Single tokens ($L = 1$) and minimal pairs ($L = 2$) lack sufficient spectral resolution for the harmonic filters to operate, causing generation collapse. Longer sequences provide the spectral density required for phase-based filtering (see Appendix I).

filter weights ($\alpha \approx 0.74$) degrades COMET by only 3%, consistent with the mathematical invariance of phase to scalar scaling ($\angle z = \angle(z/\alpha)$). Details are provided in Appendix E.

### 5.3. Spectral Density Threshold

We tested PRISM on isolated tokens ($L = 1$) and minimal pairs ($L = 2$), using German nouns verified as single tokens under the model's tokenizer. Both ambiguous and unambiguous tokens were included.

The model fails to generate coherent output in both conditions, producing repetitive loops regardless of the input token's semantic properties. This failure is predicted by the architecture: a single token produces only a DC component under FFT, and a token pair produces only two frequency bins (the Nyquist limit). In both cases, the spectral resolution is insufficient for the harmonic filters to distinguish signal from noise.

This result establishes a lower bound on the sequence length

required for phase-based computation. The model possesses the correct token identity (evidenced by the relevant tokens appearing in the repetitive output) but lacks sufficient spectral density to drive generation. This is consistent with the spectral density prediction and explains the architecture's strong performance at longer sequence lengths (Section 6.3): phase-based filtering improves as spectral resolution increases with $L$.

Together, these WMT14 results serve as hypothesis-guided observational probes rather than the main causal evidence; they motivate the WikiText-103 encoder experiments and ablations in Section 6, where phase dependence is tested directly.

## 6. WikiText-103 Experiments

We evaluate PRISM and its hybrid variants on WikiText-103 (Merity et al., 2017) at sequence length $L = 4096$ to test whether the phase-based spectral branch provides complementary value alongside attention at scale.

### 6.1. Experimental Setup

Models were trained on Wikitext-103 (Merity et al., 2017) using a BERT-style MLM objective (Devlin et al., 2019) with dynamic masking (Liu et al., 2019) and a Pre-LN topology (Nguyen & Salazar, 2019). PRISM variants utilize RMSNorm (Zhang & Sennrich, 2019), while baselines use standard LayerNorm (Ba et al., 2016). All runs used fixed seeds for exact evaluation parity.

**Optimization:** Models were trained for 40 epochs with a global batch size of 32 (8 physical × 4 gradient accumulation steps). We utilized a cosine learning rate schedule with a peak of $1 \cdot 10^{-3}$ and a 10% warmup.

**Regularization:** For the Transformer and FNet baselines, as well as the HSSM architecture, we applied standard weight decay ($\lambda = 0.01$). For the PRISM Hybrid, we set weight decay to 0.0, as the unit-norm constraint within the spectral branch provides implicit regularization. Standard dropout ($p = 0.1$) was applied to all models.

### 6.2. Architectural Designs

To isolate the contribution of Phase Coding, we evaluated four distinct architectural topologies, standardized to a ≈ 33M parameter budget. All models used 32k vocab BPE tokenizer. The hybrid architectures are illustrated in Figure 5.

**PRISM Encoder Specification:** Both experimental configurations utilize an upgraded encoder designed for generative scaling. To enable extrapolation to $L = 4096$, we replace static filters with implicit **Neural Filters**: a small MLP (3 layers, hidden dim 64, SiLU activations) that maps sinusoidal position encodings to complex-valued

filter weights $\mathbf{H} \in \mathbb{C}^{L \times d}$. Given normalized positions $t \in [0,1]^L$, the MLP outputs a complex weight for each frequency bin and feature dimension. Because the filter is implicitly parameterized rather than stored as a fixed tensor, it generalizes to arbitrary sequence lengths at inference without retraining. Crucially, we introduce **Dynamic RoSE**, which injects a content-dependent phase shift $\phi_{steer}$ alongside the positional rotation $\theta_{pos}$, formulated as $E(x_t) = z_t \cdot e^{i\theta_{pos}} \cdot e^{i\phi_{steer}}$.

**1. Transformer:** A standard Transformer encoder (Vaswani et al., 2017) with Rotary Position Embeddings (RoPE) (Su et al., 2024) and quadratic Self-Attention.

**2. FNet Hybrid:** An FNet encoder (Lee-Thorp et al., 2022) mirroring the PRISM Hybrid topology, using real-valued 2D Fourier mixing instead of complex-valued. It includes the same 1-layer Transformer refiner as the experimental models, ensuring that any performance gap stems from the mixing representation (real vs. complex) rather than the presence of attention.

**3. PRISM Hybrid** (Figure 5b)**:** Dynamic RoSE produces a complex wave output and a real particle output. The wave passes through a PRISM encoder ($d = 512$, 5 layers) and a phase bridge ($\mathbb{C} \rightarrow \mathbb{R}$); the particle bypasses the encoder. Both are concatenated and projected to $d = 512$ via a learned projection with GELU, then refined by a 1-layer Transformer. All PRISM variants use RMS-based phase normalization (Zhang & Sennrich, 2019).

**4. Hybrid Spectral Sequence Model (HSSM):** A dual-stream architecture projecting the embedding to $d = 256$ for two parallel streams: a 9-layer FNet encoder for real-valued mixing and a 9-layer PRISM encoder for complex-valued spectral filtering, fused before a shared refiner.

**5. Wave-Particle Transformer (WPT)** (Figure 5a)**:** Replaces the HSSM's FNet stream with a lightweight Transformer encoder ($L = 6$, $d = 256$) using RoPE. This stream processes token identity in parallel with the phase-coded relational stream ($L = 6$), fusing their outputs via concatenation and a 1-layer attentive refiner.

### 6.3. Results & Analysis

We compare these architectures, all normalized to approximately 33M parameters.

**Results.** As shown in Table 3, the real-valued spectral baseline (FNet-Hybrid) collapses at long contexts (9.87 PPL), suggesting that additive mixing without phase is insufficient in this setup when dynamic content selection is required. The PRISM Hybrid (6.06 PPL) substantially closes the gap using a single refiner layer. The WPT achieves the best performance (4.94 PPL), outperforming the Transformer (5.28) while using 18% fewer core parameters (12.9M vs 15.8M).

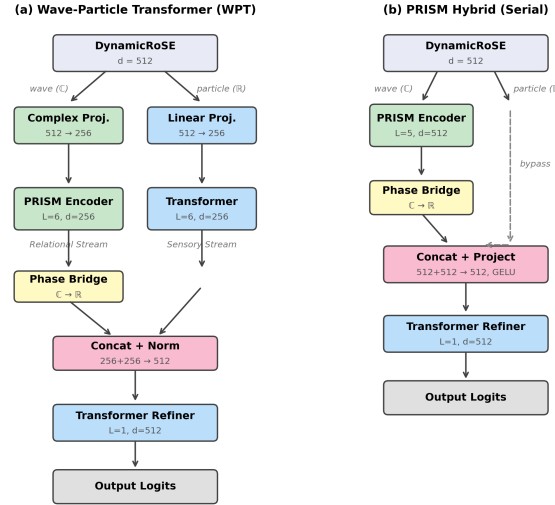

*Figure 5.* Hybrid architecture designs. **(a)** The WPT processes relational (PRISM) and sensory (Transformer) streams in parallel at $d = 256$ before fusion. **(b)** The PRISM Hybrid processes the wave through PRISM at full width ($d = 512$); the raw particle embedding bypasses the encoder and is concatenated before refinement.

*Table 3.* **WikiText-103 Masked Reconstruction Benchmark ($L = 4096$).** Comparison of architectures. The Dual-Stream WPT outperforms the Transformer baseline with fewer core parameters.

| MODEL | PARAMETERS (M) | | | PERFORMANCE | | |
|---|---|---|---|---|---|---|
| | TOT. | NO-EMB | CORE | PPL ↓ | TOP1 | TOP5 |
| *Baselines* | | | | | | |
| FNET (H) | 34.7 | 15.8 | 15.8 | 9.87 | 56.1 | 74.2 |
| TRANSF. | 32.5 | 15.8 | 15.8 | 5.28 | 66.1 | 81.9 |
| *Our Architectures* | | | | | | |
| HSSM | 33.0 | 15.1 | 13.0 | 6.47 | 63.2 | 79.5 |
| PRISM (H) | 32.9 | 16.2 | 14.0 | 6.06 | 64.3 | 80.3 |
| **WPT** | **31.8** | **15.0** | **12.9** | **4.94** | **67.2** | **82.8** |

This suggests that decoupling the spectral relational stream from the attention stream yields a more parameter-efficient architecture.

### 6.4. Interventional Ablations

The results above show that hybrid architectures with PRISM outperform baselines, but do not isolate whether the gains come from phase specifically. We performed three sets of interventional ablations to test this.

**Filter interventions.** We intervened on the learned spectral filter $\mathbf{H}$ in the FFT→$\mathbf{H}$→IFFT pipeline, modifying a single variable while keeping all other weights frozen at the trained checkpoint. Table 4 (top) reports the degradation factor relative to the baseline. Phase-only filters retain >95% of performance. Removing, reversing, or shuffling phase causes catastrophic degradation. This holds identically across two independently trained models with differ-

ent activations (ModReLU vs. CReLU) and normalization (PPLN vs. LayerNorm), consistent with the dependence on phase arising from the spectral topology, not from specific architectural constraints. The WMT14 attenuation stress test in Appendix E provides a complementary check: uniformly reducing spectral-filter magnitude while preserving phase retains most translation quality.

*Table 4.* Interventional ablations. *Top:* Filter interventions on frozen checkpoints (degradation factor vs. unmodified). Phase-only filters retain >95% of performance; removing phase is catastrophic. *Bottom:* Architecture isolation. The real→complex gap and WPT component swap isolate the contribution of phase.

| Filter interventions (same checkpoint) | | |
| CONDITION | CONSTRAINED | UNCONSTRAINED |
|---|---|---|
| $\mathbf{H} = 1$ (NO FILTER) | 328× | 78× |
| $|\mathbf{H}|$ (MAG. ONLY) | 153× | 173× |
| $\mathbf{H}/|\mathbf{H}|$ (PHASE ONLY) | 1.41× | 1.51× |
| $\mathbf{H}^*$ (PHASE REVERSED) | 245× | 139× |
| PHASE SHUFFLED | 160× | 161× |
| *Architecture isolation* | | |
| MODEL | CONFIG. | PPL ↓ |
| REAL BASELINE | SiLU + LN | 6.15 |
| PRISM (CONSTR.) | MODRELU + PPLN | 6.06 |
| PRISM (UNCONSTR.) | CRELU + LN | 5.80 |
| WPT (REAL STREAM) | 13.5M PARAMS | 5.55 |
| WPT (CRELU STREAM) | 12.9M PARAMS | 5.46 |
| WPT (COMPLEX) | 12.9M PARAMS | **4.94** |

**Real vs. complex isolation.** To test whether persistent complex representations across layers drive the gains, we trained topology-matched variants (Table 4, bottom). A real-valued spectral encoder using `rfft`/`irfft` with an expand-compress design ($d \to 2d \to d$) to match parameter capacity achieves 6.15 PPL. The constrained PRISM (ModReLU + PPLN) achieves 6.06. Notably, replacing all phase-preserving components with phase-destroying alternatives (CReLU + standard LayerNorm) *improves* standalone performance to 5.80, indicating that phase coding emerges from the complex spectral topology rather than from our architectural constraints. To test this, we performed layer-wise causal interventions on the unconstrained model: scrambling phase at any single layer causes 72–205× degradation, while scrambling magnitude causes only 1.1×. The constrained model shows even stronger phase dependence (∼2000×). This establishes that both models rely on phase as the primary information carrier, regardless of whether the activation function explicitly preserves it. Analysis of CReLU's quadrant routing supports the interpretation that the network learned content-dependent spectral gating rather than static suppression (Appendix F).

**WPT component swap.** Replacing WPT's PRISM stream with the parameter-matched real spectral encoder (13.5M core params vs. 12.9M) yields 5.55 PPL vs. 4.94 for WPT— a 12.3% gap attributable to complex phase coding in the relational stream alone. We also tested CReLU-WPT (CReLU + LayerNorm in the relational stream): 5.46 PPL, worse than

the constrained WPT (4.94) despite CReLU being better standalone (5.80 vs. 6.06). We hypothesize that CReLU's quadrant routing can suppress noise independently, reducing pressure to align phase angles precisely. In a hybrid where attention already handles token selection, ModReLU's constraint forces the phase stream to specialize in relational interference, which appears more complementary. This supports the relational primitive framing, though we present it as a hypothesis.

### 6.5. Throughput Analysis

To evaluate whether the $\mathcal{O}(N \log N)$ complexity translates into favorable scaling, we measured end-to-end training throughput (forward + backward + optimizer) on an A100 GPU. The comparison has two implementation confounds: Flash Attention provides fused CUDA kernels for the Transformer, and PyTorch currently keeps complex FFT/IFFT operations in FP32 under BF16 autocast because complex BF16 kernels are not available. Table 5 therefore reports three settings: realistic BF16 deployment, FP32 parity, and FP32 with Flash Attention disabled.

*Table 5.* Training throughput (tokens/s, batch=2) on A100. *Top:* BF16 autocast reflects realistic deployment with Flash Attention enabled. *Middle:* FP32 parity removes the BF16 precision confound. *Bottom:* FP32 without Flash Attention removes both precision and fused-attention confounds; dashes denote settings not run or not applicable after OOM. Under FP32 parity, PRISM retains 77% of throughput from $L$=4096 to $L$=16384, vs. the Transformer's 35%.

| BF16 autocast, Flash Attention on | | | |
| MODEL | 4096 | 8192 | 16384 | MEM (GB) |
|---|---|---|---|---|
| TRANSF. | 237,248 | 215,703 | 166,419 | 16.6 |
| FNET | 287,787 | 300,555 | 288,252 | 16.1 |
| PRISM | 118,793 | 133,416 | 132,226 | 23.9 |
| WPT | 95,341 | 119,809 | 103,376 | 23.3 |
| FP32 parity, Flash Attention on | | | |
| MODEL | 4096 | 8192 | 16384 | MEM (GB) |
| TRANSF. | 90,486 | 57,209 | 31,970 | 22.2 |
| FNET | 178,368 | 148,736 | 105,378 | 22.1 |
| PRISM | 91,602 | 87,861 | 70,857 | 28.4 |
| WPT | 63,956 | 47,677 | 30,013 | 29.7 |
| FP32, Flash Attention off | | | |
| MODEL | 4096 | 8192 | 16384 | MEM (GB) |
| TRANSF. | 70,796 | OOM | – | – |
| FNET | 161,212 | 115,581 | – | – |
| PRISM | 86,946 | 72,954 | – | – |
| WPT | 56,341 | OOM | – | – |

The BF16 throughput gap (237K vs. 119K tok/s at $L$=4096) largely reflects current framework support: real-valued attention benefits from fused Flash Attention and native BF16, whereas complex FFT/IFFT remains in FP32. Removing the BF16 confound narrows the comparison: at FP32 parity, PRISM matches the Transformer's throughput at $L$=4096

and degrades more gracefully with sequence length, retaining 77% of its throughput from $L$=4096 to $L$=16384 while the Transformer retains 35%. Disabling Flash Attention further removes the fused-kernel confound; in this setting, the Transformer and WPT run out of memory at $L$=8192, while PRISM continues to operate (86.9K tok/s at $L$=4096, 73.0K at $L$=8192). These results separate absolute throughput, which is currently framework-limited for complex-valued models, from intrinsic sequence-length scaling, which follows the expected $\mathcal{O}(N \log N)$ vs. $\mathcal{O}(N^2)$ trend.

# 7. Discussion & Conclusion

This work set out to investigate the computational role of phase in neural sequence modeling. We summarize the key findings.

## 7.1. The Model Prefers Complex Representations

A topology-matched real-valued spectral encoder, using `rfft`/`irfft` with a $d \to 2d \to d$ expand-compress design, achieves 6.15 PPL despite having more core parameters (13.5M) than the complex PRISM variant (12.9M, 5.80 PPL). The real encoder projects into an unconstrained 1024-dimensional space, where cross-channel interactions must be learned freely. Complex multiplication, by contrast, imposes a structured coupling between real and imaginary components:

$$(W_r x_r - W_i x_i) + i(W_r x_i + W_i x_r). \tag{14}$$

This form acts as an inductive template for rotation and interference: the subtractive cross-term is built into the parameterization rather than discovered from unconstrained real mixing. Although an unconstrained real-valued model could represent such interactions in principle, the matched real baseline does not recover them as effectively under our training setup. This suggests that the complex parameterization provides a useful bias for spectral sequence modeling.

## 7.2. Phase Carries Task-Relevant Information

Our filter ablations show that the learned spectral filters are strongly phase-dependent: retaining only the phase of the spectral filter ($\mathbf{H}/|\mathbf{H}|$) preserves >95% of performance, while retaining only the magnitude ($|\mathbf{H}|$) is catastrophic. Layer-wise activation interventions on the complex PRISM encoder state show the same pattern: scrambling phase at an individual PRISM layer degrades performance by 72–205$\times$, while scrambling magnitude causes only 1.1$\times$ degradation. This does not imply that magnitude is unused everywhere in the model; residual paths, gates, and real-valued components can still carry scale and salience information. Rather, the result shows that the learned complex spectral pathway is highly phase-dependent, with phase carrying the dominant task-relevant relational signal. The importance of

phase in structured representations has precedent in signal processing (Oppenheim & Lim, 1981); our results extend this observation to learned neural sequence representations (Appendix C).

## 7.3. Emergent, Not Prescribed

Unlike Holographic Transformers (Huang et al., 2025), which prescribe interference via explicit phase decay and coherent superposition, PRISM provides only the spectral topology (FFT→H→IFFT) and complex linear algebra. Phase-based computation emerges from training. The CReLU experiment is the strongest evidence for this: replacing all phase-preserving components (ModReLU, PPLN) with phase-destroying alternatives *improves* standalone performance, yet the model still organizes computation around phase. The network discovers that the complex spectral pathway is an efficient substrate for relational structure, complementing attention's role in token selection.

**Scope of mechanistic claims.** Our analysis differs from circuit-level mechanistic interpretability (Nanda et al., 2023; Olsson et al., 2022) in both granularity and methodology. We do not localize computation to specific attention heads or reverse-engineer the algorithm a trained network discovered. Instead, we provide interventional evidence at the level of representational properties: layer-wise phase scrambling and spectral filter ablations causally establish that phase, not magnitude, carries the relational signal. We adopt the term "mechanistic" in the broader sense of identifying causal mechanisms via intervention, rather than the narrower sense of full algorithmic reverse-engineering.

## 7.4. Limitations & Future Directions

Our goal is to isolate the causal role of phase in spectral neural sequence models, using language as a structured sequence-domain workbench. The results show that phase can emerge as a dominant information carrier: preserving phase retains most performance, while removing, reversing, or shuffling it causes severe degradation, even when explicit phase-preserving constraints are weakened. This is a mechanistic study, not a foundation-model benchmark. Future work should test whether similar phase-based relational structure appears in other architectures, domains, objectives, and end-to-end complex-valued decoders. PRISM is also a computational model, not an optical hardware proposal; physical realization remains future work.

To facilitate further investigation, we release the implementation; see Code Availability.

# Acknowledgements

We used large language models as assistive tools for generating figures, polishing manuscript text, and routine coding

tasks. All scientific contributions, experimental design, and architectural decisions are solely the work of the authors.

## Impact Statement

This paper presents work whose goal is to advance the field of Machine Learning, specifically by understanding the computational role of phase and wave mechanics in neural networks. While there are many potential societal consequences of advancing deep learning and sequence modeling broadly, there are no specific consequences of this fundamental mechanistic investigation that we feel must be highlighted here.

## Author Contributions

A.Y. conceived the project, designed the PRISM architecture, implemented the codebase, conducted all experiments and ablations, and wrote the manuscript. İ.Y. provided academic supervision and guidance throughout the project.

## Code Availability

Code is available on GitHub at `https://github.com/AlperYildirim1/Language-as-Waves`.

Trained checkpoints are available through Hugging Face at `https://huggingface.co/prism-lab`.

An archival snapshot is available on Zenodo at `https://doi.org/10.5281/zenodo.17330341`.

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

## A. Phase Coherence: Antonym Analysis

**Antonym coherence and BERT sanity check.** One might expect antonyms to exhibit anti-phase (negative coherence) rather than positive coherence. However, computing phase coherence without the absolute value in Eq. 13 shows that antonyms still exhibit positive coherence (mean $R = 0.125$, 95% CI $[0.104, 0.149]$), not anti-phase. We ran BERT as a sanity check and observed similar behavior: antonyms are encoded near each other in embedding space. This is expected from distributional semantics—"cold" and "hot" appear in nearly identical contexts ("the tea is hot/cold"), so embeddings encode them in shared spectral neighborhoods. Phase coherence thus reflects *semantic relatedness* (topic binding) rather than polarity, grouping semantically related concepts regardless of their logical relationship.

## B. Computational Efficiency Data

We analyze the relationship between gate activity and phase rotation across layers in the PRISM encoder. Gate openness measures the mean activation of the gating mechanism (Eq. 3), representing the computational effort expended by each layer. Phase rotation ($\Delta\theta$) measures the angular displacement applied to token representations.

Figure 6 shows that early layers have high gate activity but low phase rotation, while deeper layers clamp their gates while producing large rotations. We quantify this as the ratio $\eta = \Delta\theta/$gate openness (Table 6).

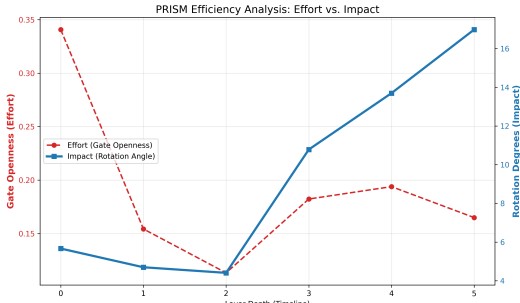

*Figure 6.* Gate openness (red) vs. phase rotation (blue) by layer. Early layers expend high gate activity for modest rotations; deeper layers produce large rotations with minimal gating.

*Table 6.* Efficiency ratio $\eta$ (phase rotation / gate openness) by layer.

| Layer | $\eta$ |
|---|---|
| 0 | 16.62 |
| 1 | 30.44 |
| 2 | 38.91 |
| 3 | 59.15 |
| 4 | 70.65 |
| **5** | **102.98** |

## C. Phase vs. Magnitude in Learned Representations

The primacy of phase over magnitude in signal reconstruction was established by Oppenheim & Lim (1981), who showed that swapping the phase spectrum of one image with another produces a result perceptually dominated by the phase donor. Our filter ablations (Table 4) demonstrate an analogous phenomenon in learned neural representations: retaining only the phase of the spectral filter preserves nearly all task performance, while retaining only the magnitude is catastrophic.

We hypothesize that magnitude is inherently less stable as an information carrier in deep networks. Under the spectral filtering operation $Y = \text{IFFT}(\text{FFT}(X) \odot H)$, the magnitude $|H|$ scales energy but does not create structured interference patterns. Phase, by contrast, determines *where* constructive and destructive interference occurs across frequency bands, enabling selective suppression and amplification of spectral components. In a deep stack of such operations, magnitude scaling compounds multiplicatively (risking explosion or vanishing), while phase rotations compose geometrically on the unit circle, providing a naturally bounded and stable substrate for information encoding.

## D. Complex-Valued Causal Decoder

A natural extension of PRISM is an end-to-end complex-valued architecture with a causal decoder. We investigated this and concluded it is not practical with the current design. The Gated Harmonic Convolution performs circular convolution via FFT→H→IFFT, where every output position depends on all input positions simultaneously. There is no way to apply a causal mask after IFFT, because future information is already mixed into the result. Enforcing causality in the frequency domain would require constraining $\text{IFFT}(H)$ to be zero for negative indices—a nontrivial spectral constraint on the MLP-parameterized filter. Even if solved for parallel training, autoregressive inference would require recomputing the full FFT for each new token, giving $\mathcal{O}(L^2 \log L)$ total cost. A causal reparameterization of the spectral filter or a recurrent formulation of the spectral convolution remain open research directions.

## E. Scalar Attenuation Robustness

To test whether the WMT14 model is robust to reductions in spectral-filter magnitude, we applied a hybrid hardware-compiler perturbation to the trained checkpoint. All learned spectral filters $\mathbf{H}$ in the PRISM encoder were uniformly attenuated by a scalar factor ($\alpha \approx 0.74$), bringing their maximum modulus below one, while the digital gating, mixing, bridge, and decoder layers retained their learned gain. In the reported stress test, we additionally injected small static per-

turbations into the optical filters, and weaker perturbations into digital pointwise weights, to approximate hardware noise. Under this perturbation, PRISM retained 97% of its baseline COMET score (0.775 vs. 0.799), consistent with the WikiText filter ablations in Section 6.4: phase-only or phase-preserving interventions are far less damaging than phase removal, reversal, or shuffling. This supports the interpretation that much of the semantic information in the spectral branch is carried by phase, which is invariant to positive scalar attenuation: $\angle(\alpha z) = \angle z$ for $\alpha > 0$.

## F. CReLU Quadrant Gating Analysis

CReLU applies ReLU independently to real and imaginary components: $\text{CReLU}(z) = \text{ReLU}(\text{Re}(z)) + i\,\text{ReLU}(\text{Im}(z))$. This zeros quadrant Q3 ($\text{Re} \leq 0, \text{Im} \leq 0$), snaps Q2/Q4 to the real or imaginary axis, and passes Q1 unchanged. Unlike ModReLU, which preserves phase by construction, CReLU can destroy phase information.

To test whether CReLU acts as a static or dynamic gate, we tracked the fraction of dimensions falling into Q3 (the zeroed quadrant) per layer across the test set. If CReLU were indiscriminately destroying phase, the Q3 frequency would be uniform across layers and dimensions. Instead, Table 7 shows layer-specific routing patterns:

*Table 7.* Per-layer distribution of Q3 frequency across dimensions in the unconstrained (CReLU) model. Each row shows what fraction of dimensions fall into each Q3-frequency bucket. The non-uniform distribution indicates content-dependent spectral gating.

| LAYER | 0–20% | 20–40% | 40–60% | 60–80% | 80–100% |
|---|---|---|---|---|---|
| 0 | 0.0 | 15.0 | 81.1 | 3.9 | 0.0 |
| 1 | 0.0 | 0.0 | 6.1 | 77.9 | 16.0 |
| 2 | 0.0 | 2.3 | 28.1 | 57.8 | 11.7 |
| 3 | 0.4 | 7.8 | 46.5 | 39.1 | 6.2 |
| 4 | 0.8 | 23.4 | 55.3 | 19.3 | 1.2 |

Layer 1 pushes 78% of dimensions into heavy Q3 gating (60–80% frequency), while Layer 4 spreads dimensions more evenly with 23% in the 20–40% bucket. The same dimensions are not consistently pushed to the same quadrant—the routing is input-dependent. This indicates that the network exploits CReLU's quadrant geometry as a learned, content-dependent spectral gate, selectively suppressing dimensions via Q3 while preserving phase information in Q1.

## G. Baseline Selection

Our experimental design prioritizes variable isolation over benchmark performance. To test whether phase encoding provides representational advantages, we compare against a baseline that is topologically identical but operates in the real-valued domain.

**Why FNet.** Both FNet and PRISM are global spectral mixers operating at $\mathcal{O}(N \log N)$ complexity. The key difference is that FNet applies a real-valued Fourier transform ($T_{\text{FNet}}(x) = \text{Re}(\mathcal{F}(x))$), while PRISM applies complex-valued spectral filtering with learnable phase. This ensures that performance differences are attributable to the representation type (real-valued vs. complex-valued) rather than the mixing mechanism.

**Why not Mamba/SSMs.** Mamba (Gu & Dao, 2024) is a strong efficient baseline, but it introduces two confounding variables: recurrence (state-space memory) and data-dependent gating on the time axis. Comparing PRISM to Mamba would conflate the effects of complex-valued representation with the effects of recurrence, making it impossible to attribute performance differences to phase encoding specifically. We therefore restrict comparisons to architectures that share PRISM's global spectral mixing topology.

## H. Optical Correlator Analogy

The gated harmonic convolution (Eq. 4) is structurally analogous to an optical 4f-correlator (Chi & George, 2011), in which an input field is Fourier-transformed by a lens, multiplied by a filter in the focal plane, and inverse-transformed by a second lens. In such a system, the filtering operation occurs via wave propagation at effectively $\mathcal{O}(1)$ latency with respect to input size.

We note this analogy as a theoretical property of the computational structure: architectures that perform filtering exclusively through phase manipulation (without gain) are in principle compatible with passive optical implementation, where no active amplification is required. This does not constitute a claim that PRISM is hardware-ready, but rather that the class of phase-only spectral architectures has a natural physical realization.

## I. N-Slit Diffraction Analogy

The spectral density threshold observed in Section 5.3 can be understood through an analogy to N-slit diffraction in classical optics.

### I.1. Physical Correspondence

In the N-slit experiment, a coherent plane wave illuminates $N$ equally-spaced slits. Each slit acts as a secondary source, and the resulting wavefronts interfere at a distant screen. The intensity pattern is:

$$I(\theta) = I_0 \left| \sum_{n=1}^{N} e^{in\delta} \right|^2 = I_0 \frac{\sin^2(N\delta/2)}{\sin^2(\delta/2)} \qquad (15)$$

where $\delta = \frac{2\pi d \sin \theta}{\lambda}$ is the phase difference between adjacent slits.

We propose the following correspondence:

| Optics | PRISM |
|---|---|
| Coherent light source | Shared embedding space |
| Slit apertures | Token positions |
| Slit count $N$ | Sequence length $L$ |
| Phase delay $\delta$ | Positional phase shift $\omega t$ |
| Interference pattern | Filtered representation |

### I.2. Resolution Limits

This analogy maps directly to the observed failure modes:

At $N = 1$ (single slit), no interference occurs—the output is a broad, featureless diffraction envelope. In PRISM, $L = 1$ produces only a DC component under FFT, providing no spectral structure for the harmonic filters.

At $N = 2$ (double slit), basic sinusoidal fringes emerge but angular resolution is poor. In PRISM, $L = 2$ yields only two frequency bins (the Nyquist limit), insufficient for the filters to selectively pass or suppress frequency bands.

At $N \gg 1$ (diffraction grating), sharp principal maxima appear with resolution scaling as $\Delta\theta \propto 1/N$. In PRISM, longer sequences provide denser spectral sampling, enabling precise filtering. This is consistent with PRISM's strong performance at $L = 4096$ (Section 6.3).

## J. Relation to Complex-Valued State Space Models

Concurrent work on Mamba-3 (Lahoti et al., 2026) also studies the value of complex-valued dynamics in sequence models. Its motivation is different from ours: Mamba-3 introduces complex-valued state updates in a recurrent state-space model to improve state tracking, while PRISM uses a complex-valued spectral pathway to study whether phase becomes a task-relevant information carrier inside a trained network.

The overlap is therefore conceptual rather than architectural. Both works suggest that complex-valued representations are useful for expressing rotational or phase-like structure across sequences. PRISM contributes complementary interventional evidence: preserving phase in the spectral pathway retains most performance, while disrupting phase causes severe degradation. Understanding when recurrent complex state updates are sufficient, and when persistent complex spectral representations are beneficial, remains an open direction.

