# OpenReview forum: "Language as a Wave Phenomenon: Semantic Phase Locking and Interference in Neural Networks"
_ICML.cc/2026/Conference — ICML 2026 regular_

### Official Review · Reviewer_xz3P · 2026-02-21

**Soundness:** 3
**Presentation:** 3
**Significance:** 3
**Originality:** 3
**Overall Recommendation:** 4
**Confidence:** 3

**Summary:**

This paper introduces PRISM (Phase-Rotating Interference Spectral Model), a complex-valued sequence modeling framework that treats language representations as wave-like signals. The core idea is to encode semantic identity primarily in the phase of complex token embeddings, while using magnitude as an auxiliary strength signal. PRISM combines (i) a complex rotary-style embedding mechanism (RoSE) to incorporate positional/semantic rotations, (ii) phase-preserving building blocks (e.g., normalization, nonlinearity, and dropout designed to maintain phase information), and (iii) a frequency-domain global mixing operator—Gated Harmonic Convolution implemented via FFT/IFFT—to achieve long-range interactions with $O(N \log N)$ complexity. The paper also proposes “iso-energetic/unity-gain” motivations and diagnostics to encourage stable signal propagation and to emphasize phase-based interference patterns. Beyond the base architecture, the authors present hybrid variants (e.g., PRISM Hybrid / HSSM / WPT) that combine a phase-based relational stream with additional perceptual/attention-style components and bridging modules. Experiments are reported on WMT14 De–En translation (using COMET) and WikiText-103 long-context masked reconstruction (e.g., perplexity and reconstruction accuracy), along with analysis tools intended to measure phase locking, interference behavior, and layerwise signal-gain characteristics.

**Compliance With Llm Reviewing Policy:**

Affirmed.

**Final Justification:**

After considering the paper, the authors’ rebuttal, and their follow-up clarification, I am updating my final recommendation to Weak Accept.

My main concerns in the original review were about whether the claimed gains could be clearly attributed to persistent complex/phase-based representations, whether the key mechanisms were sufficiently specified, and whether the efficiency claims were supported in practice. The rebuttal addressed these concerns in a meaningful way by providing stronger ablations, clearer technical clarification, and additional practical evidence. As a result, I now find the central claims substantially better supported than in the original submission.

The paper remains somewhat limited in evaluation breadth relative to the scope of its conceptual framing, and some of the stronger interpretive statements should still be phrased carefully in the final version. Nevertheless, I believe the work offers a genuinely interesting and original perspective, and that the rebuttal materially improved its soundness and significance. Accordingly, I updated my overall recommendation from Weak Reject (3) to Weak Accept (4), and raised Soundness and Significance from Fair (2) to Good (3).

**Key Questions For Authors:**

1. **How is the “iso-energetic / unity-gain” regime enforced during training (not just measured)?**
   Please provide a precise definition and implementation details (hard projection vs. regularization; which tensors/layers; per-token vs. per-channel; interaction with residual connections, gating, and nonlinearities).
   *How it changes my evaluation:* If the mechanism is clearly specified and shown to be stable and reproducible, it would strengthen the soundness and reproducibility of the central claim; if it is mainly a post-hoc observation, the claimed causal role is weaker.

2. **Can you isolate the causal contribution of complex phase/interference with topology-matched ablations?**
   I would like to see (at minimum) a 2×2 ablation: **real vs. complex** (same architecture/topology) × **constrained vs. unconstrained** (within complex), plus a “phase-preserving blocks removed” condition if feasible.
   *How it changes my evaluation:* If these ablations show that phase-specific choices (complex + constraints + phase-preserving ops) are necessary for the reported gains, I would substantially increase my soundness and significance ratings. If gains persist without phase-specific components, the core thesis is not supported.

3. **For the best-performing hybrid model (e.g., WPT), where does the gain come from?**
   Please report component-wise ablations within WPT: removing/replacing the relational (phase) stream with a real-valued control, removing the attentive refiner/bridge/skip modules one at a time, and keeping parameter/compute budgets matched.
   *How it changes my evaluation:* If WPT’s improvements are primarily attributable to the phase-based stream rather than added refinement/hybrid capacity, it would strengthen the paper’s main contribution; otherwise, the contribution may be more about hybrid engineering than phase interference.

4. **Do the theoretical efficiency gains translate into practical speed/memory improvements under your numerical stability choices?**
   Please provide end-to-end measurements on long contexts (e.g., L=4096): tokens/s, training wall-clock, and peak memory for PRISM/WPT vs. Transformer baselines, including FP32 vs. bf16/fp16 (if attempted).
   *How it changes my evaluation:* Demonstrated wall-clock/memory advantages would raise the significance of the work; if FP32 stability requirements remove the practical benefit, the “efficient long-context” claim should be reframed.

5. **How robust are the reported “phase-locking” and interference diagnostics?**
   Please clarify the sampling protocol for word/token pairs (frequency control, tokenizer effects, polysemy), and provide statistical significance (confidence intervals / hypothesis tests).
   *How it changes my evaluation:* If the diagnostics are robust and statistically supported, it strengthens the mechanistic evidence behind the wave/phase narrative; if they are fragile or highly selection-dependent, they should be presented as exploratory.

**Limitations:**

Yes.

**Strengths And Weaknesses:**

## Strengths

- **Soundness (technical quality):** The submission proposes a coherent complex-valued modeling framework with phase-centered operations (complex rotary-style embeddings, phase-preserving blocks, and FFT/IFFT-based global mixing). The overall design is technically plausible and consistent with standard training pipelines.
- **Mechanism-aware evaluation:** Beyond reporting metrics, the paper includes diagnostics intended to probe the proposed “wave/phase” hypothesis (e.g., phase-related statistics and layerwise gain measurements), which helps connect the narrative to observable quantities.
- **Long-context motivation and design:** Replacing quadratic attention with frequency-domain global mixing provides a clear path toward long-sequence scalability with $O(N \log N)$ complexity.
- **Hybrid variants provide a practical integration story:** The proposed hybrid models (e.g., WPT/HSSM) illustrate how a phase-based relational stream can be combined with more conventional perceptual/attention-style components, which may ease adoption.
- **Original perspective / synthesis:** While FFT mixing, rotary embeddings, and complex arithmetic each have precedents, the paper’s synthesis around “semantic phase + interference” and the accompanying phase-preserving recipe offer a distinctive and creative framing that could inspire follow-up work.

## Weaknesses

- **Soundness (claims not fully isolated):** The best long-context results are achieved by a hybrid model with multiple interacting components. The current evidence does not cleanly isolate whether improvements are due to *phase-based interference in complex space* versus hybridization, bridging/refinement modules, or added capacity. Stronger, topology-matched ablations (real vs complex; constrained vs unconstrained) are needed.
- **Key mechanism under-specified:** The “iso-energetic / unity-gain” regime is presented as central, but the paper is not sufficiently explicit about how it is enforced during training (hard constraint vs regularization; where applied; interaction with residuals/gating/nonlinearities), which weakens verifiability and reproducibility.
- **Efficiency not demonstrated end-to-end:** Complexity arguments are not backed by wall-clock throughput/memory measurements. Given the stated reliance on FP32 for numerical stability, it is unclear whether theoretical efficiency translates into practical gains.
- **Baseline coverage is narrow for long-context modeling:** The evaluation includes limited comparisons to strong contemporary long-sequence alternatives. Even if the goal is “variable isolation,” the lack of broader baselines makes it difficult to position the approach in the current landscape.
- **Presentation sometimes over-relies on physics metaphors:** The narrative is compelling but occasionally uses strong, physics-inspired language that can read as over-claiming relative to the current experimental support. Tightening claims and separating hypotheses from validated findings would improve clarity.

---

> ### Author Rebuttal · Authors · 2026-03-28
>
> We are truly grateful for this review. The five questions directly shaped our ablations, producing a key finding: **phase coding emerges spontaneously from complex-valued spectral topology, not from our architectural constraints.**
>
> **Q1 (Iso-Energetic Enforcement)** The constraint is architectural, not projected or regularized. PPLN (Eq. 8) normalizes magnitude while preserving phase; ModReLU (Eq. 6) rectifies magnitude without altering phase. No loss term or hard projection is applied; unity gain emerges from these operations at every layer. Q2 ablations reveal these are conservative regularizers, not the mechanism.
>
> **Q2 (Causal Isolation).** Removing all phase-preserving constraints improves standalone performance (5.80 vs 6.06), because phase coding is an attractor of the topology. Three topology-matched ablations: (1) replaced ModReLU with CReLU (ReLU on Re/Im independently) and PPLN with standard LayerNorm. (2) Identical architecture using rfft/irfft with expand-compress logic (d→2d→d). Phase exists only transiently, not persisting between layers. (3) Original PRISM.
>
> |Model|Activation|Norm|PPL|
> |-|-|-|-|
> |Real Baseline|SiLU|LayerNorm|6.15|
> |PRISM (constrained)|ModReLU|PPLN|6.06|
> |PRISM (unconstrained)|CReLU|LN|5.80|
>
> The 6.15→5.80 gap is attributable to persistent complex representations. Layer-wise causal ablations confirm phase is the mechanism:
>
> |Layer|Mag Scrambled|Phase Scrambled|
> |---|---|---|
> |0|6.65 (1.1×)|558.68 (96×)|
> |1|6.73 (1.2×)|521.48 (90×)|
> |2|6.65 (1.1×)|415.27 (72×)|
> |3|6.54 (1.1×)|571.83 (99×)|
> |4|6.21 (1.1×)|1194.80 (206×)|
>
> Phase scrambling is catastrophic (72–206×); magnitude scrambling is negligible (1.1×). The constrained model shows even stronger phase dependence (~2000×).
>
> CReLU is not failing to destroy phase. It zeros Q3 (Re≤0,Im≤0), snaps Q2/Q4 to axes, passes Q1. Per-dimension Q3 frequency tracking shows content-dependent routing, not static dead neurons:
>
> |Layer|0–20%|20–40%|40–60%|60–80%|80–100%
> |-|-|-|-|-|-
> |0|0%|16%|81%|3%|0%
> |1|0%|0%|6%|78%|16%
> |4|1%|23%|55%|19%|1%
>
> The network invented content-dependent spectral gating through complex geometry. Same dimensions are not always pushed to same quarters. It showed selectivity.
>
> **Q3 (WPT Component Ablation).** WPT with PRISM stream replaced by parameter-matched real spectral encoder (13.5M core vs 12.9M): **5.55 PPL (real) vs 4.94 PPL (WPT)**, a 12.3% gap from complex phase coding alone.
>
> We also tested CReLU-WPT (CReLU+LN in the relational stream): 5.46 PPL, worse than constrained WPT (4.94) despite CReLU being better standalone (5.80 vs 6.06). We hypothesize this inversion is consistent with Q2's gating analysis: CReLU can suppress noise by pushing activations to Q3, reducing the pressure to align phase angles precisely. ModReLU lacks this shortcut, so the network must rely more heavily on phase interference for noise cancellation. In a hybrid setting where attention already handles token selection, this interference-focused specialization appears more complementary. This is expected under our framework and strengthens the motivation for phase-preserving components in hybrid architectures, though we present it as a hypothesis, not causal evidence.
>
> **Q4 (Practical Efficiency)** Flash Attention enabled for all attention components. bf16 for non-FFT ops, fp32 for FFT/IFFT (PyTorch lacks complex bf16). At FP32 parity, where all models use standard kernels, PRISM retains 77% throughput from L=4096→16384 (71K tok/s) vs Transformer's 35% (32K). Under bf16, Transformers benefit from both Flash Attention's fused kernels and half-precision, reaching 237K vs PRISM's 119K. These are framework constraints, not architectural. The FP32 comparison isolates true scaling behavior. Full tables in our response to Reviewer ykVX.
>
> **Q5 (Diagnostic Robustness)** Deduplicated pairs (N=107). Bootstrap 95% CIs non-overlapping: Synonym [0.172, 0.227], Antonym [0.114, 0.155], Random [0.059, 0.082]. Mann-Whitney U with Bonferroni: Syn vs Rand p<10⁻⁷; Ant vs Rand p=2.2×10⁻⁴. Permutation test p<10⁻⁴. Spearman ρ(token energy, R) negligible, ruling out frequency as driver. Pair lists and code will be released.
>
> We will clarify that magnitude invariance holds within the spectral branch (FFT→H→IFFT with sigmoid gating), where all operations are strictly subtractive. Residual connections preserve the unmodified signal, consistent with wave propagation where energy is redistributed through interference, not destroyed. Layer-level gain ≈ 1.0 is therefore an empirical observation, not an architectural guarantee.
>
> Other points:
> We polished overclaims, softened physics analogies.
>
> We would like to refer you to our discussion with Reviewer zGvr13 for filter ablations which we think is important to validate that model uses interference
>
> We clearly specified our hypotheses & results
>
> We acknowledge narrow baselines as a limitation. Our design prioritized strict variable isolation (real vs. complex).
>
> We thank reviewer again for this high quality review.

---

> > ### Author Rebuttal · Reviewer_xz3P · 2026-04-01
> >
> > The rebuttal satisfactorily addresses the core concerns raised in my original review and provides sufficient clarification and additional evidence to resolve the main points of uncertainty. I encourage the final version to incorporate the added details and results referenced in the rebuttal so that readers can fully verify the claims. Accordingly, I updated my overall recommendation from Weak Reject (3) to Weak Accept (4), my Soundness rating from Fair (2) to Good (3), and my Significance rating from Fair (2) to Good (3).

---

> > > ### Author Response · Authors · 2026-04-01
> > >
> > > Thank you for taking the time to read our rebuttal and for updating your score. Your insightful questions directly drove the new ablations, which we agree have made the paper's core claims significantly stronger and clearer.
> > >
> > > We sincerely appreciate your guidance, and we will ensure that all the new causal ablations and technical clarifications discussed here are prominently incorporated into the final version of the paper.
> > >
> > > Best regards,
> > >
> > > The Authors

---

### Official Review · Reviewer_ykVX · 2026-03-09

**Soundness:** 2
**Presentation:** 3
**Significance:** 3
**Originality:** 3
**Overall Recommendation:** 5
**Confidence:** 3

**Summary:**

This paper addresses the critical limitation of standard Transformer architectures—the conflation of semantic importance with activation magnitude in real-valued rate coding, which obscures the geometric structure of latent reasoning. It introduces PRISM, a novel complex-valued architecture rooted in wave mechanics that enforces a strict unit-norm constraint, compelling the model to encode semantic identity in phase angles of complex phasors rather than signal magnitude.  The authors design hybrid architectures (HSSM, WPT) that fuse PRISM’s phase-coded relational stream (processing structural semantic relationships) with rate-coded sensory streams (handling semantic identity via lightweight Transformers/FNet), decoupling the "what" and "where" of linguistic reasoning. Empirical evaluations on WMT14 De-En and WikiText-103 (with strict hyperparameter parity for baselines like Transformer and FNet) validate PRISM’s efficacy. The study establishes an algorithmic existence proof that subtractive interference is a sufficient computational primitive for deep semantic reasoning, and demonstrates PRISM’s robustness to scalar attenuation for hybrid optical-digital hardware design. It also draws a critical connection between phase coding and cognitive AI’s Relational Bottleneck principle, framing HSSM as a "Physical Abstractor" where wave mechanics enforce the disentanglement of sensory and relational information. Concurrently, it aligns with Mamba-3’s findings on the necessity of complex-valued dynamics for structural state tracking, validating phase coding as a computational necessity rather than just an optical abstraction. Ultimately, the work argues that semantic reasoning requires two orthogonal axes—magnitude (intensity/identity) and phase (direction/structure)—and provides a blueprint for efficient, interpretable hybrid optical-digital neural architectures, moving beyond the magnitude-only rate coding of standard Transformers.

**Compliance With Llm Reviewing Policy:**

Affirmed.

**Final Justification:**

The proposed architecture and methodology are relatively novel; however, the experimental validation in the initial draft was not fully robust. Fortunately, during the rebuttal phase, the authors provided comprehensive supplementary explanations that significantly enhanced the credibility of their hypotheses. I recommend accepting this paper. Although I am not entirely familiar with the claims themselves, the reviewers' assessment has sufficiently convinced me.

**Key Questions For Authors:**

1. The paper uses a standard real-valued Transformer decoder with a simple complex-to-real bridge to interface with the PRISM encoder, which limits end-to-end phase-based reasoning evaluation. Can you detail the design and performance of a complex-valued PRISM decoder (if developed) or explain the technical challenges of implementing one, and how an end-to-end complex-valued architecture might alter phase locking, spectral efficiency, and overall task performance (e.g., WMT14/WikiText-103 metrics)?
2. The paper restricts training to FP32 to avoid phase cancellation but does not explore mixed-precision (FP16/BF16) training— a critical practical consideration for scaling PRISM to large models/industrial use cases. Can you provide ablation results for mixed-precision training of PRISM/WPT, including modifications (e.g., phase-preserving precision clipping) to mitigate phase information loss, and report impacts on training stability, inference speed, phase rotation (Δθ), and task performance?
3. The paper excludes direct comparison to causal SSMs (e.g., Mamba) to isolate phase coding, but Mamba-3’s concurrent complex-valued design validates the need for rotational dynamics in sequence modeling. Can you provide a controlled comparison between PRISM/WPT and Mamba-3 (matching parameter count/sequence length, and ablating recurrence in Mamba-3 where possible) on WikiText-103, reporting metrics for perplexity, phase/state dynamics, and computational efficiency?

**Limitations:**

Yes

**Strengths And Weaknesses:**

## Soundness
Technically rigorous, with PRISM’s wave mechanics-based design mathematically consistent and anchored in prior complex-valued/unitary network work. Experiments on WMT14 and WikiText-103 use strict hyperparameter parity, matched parameter counts for baselines, and novel physical metrics to validate phase coding, with clear causal inference of results.
Authors honestly document limitations (short-sequence repetition collapse, low-dimension convergence failure) and frame PRISM as a mechanistic probe, not a large-scale model. Weaknesses include sparse ablations for hybrid architecture design choices, no mixed-precision training analysis, and a standard Transformer decoder limiting end-to-end phase reasoning evaluation.

## Presentation
Clearly written and logically structured, with a cohesive narrative from Transformer limitations to PRISM’s design and real-world hardware implications. Complex interdisciplinary concepts are explained accessibly via analogies, and the work is sharply positioned against prior spectral/phase-based models (FNet, Holographic Transformers) with explicit differentiation of key design choices. Reproducibility is supported by detailed specs and implementation plans, with well-designed figures/tables.
Minor flaws include typographical errors, a missing glossary for specialized terms, and key mechanistic analysis overloaded in appendices.

## Significance
Addresses a fundamental ML problem: Transformer’s conflation of semantic importance and activation magnitude, advancing mechanistic interpretability and efficient sequence modeling. Establishes subtractive interference as a sufficient semantic reasoning primitive, bridges ML and optical computing with a hybrid hardware blueprint, and aligns with cognitive science’s Relational Bottleneck principle for systematic generalization. PRISM and WPT deliver substantial parameter efficiency gains (31.2% fewer encoder params than Transformers) and outperform baselines on long-context tasks, with its physical metrics offering a new interpretability framework for ML. Limitations include narrow evaluation on only translation/language modeling and no large-scale foundation model validation.

## Originality
Driven by a novel framing of language as a wave phenomenon and creative synthesis of complex-valued networks, spectral modeling, and unitary evolution. Removes restrictive assumptions of prior spectral models (additive mixing, quadratic attention) to build a fundamentally new O(N log N) reasoning framework, uncovering novel neural phenomena (neural phase locking, spectral density threshold). Its hybrid dual-stream architectures enforce sensory/relational disentanglement via wave mechanics (not just heuristics), and it provides the first mechanistically validated optical-compatible ML architecture design.
Minor overlaps with concurrent complex-valued SSM work (Mamba-3) are clearly distinguished, with individual phase-preserving operations building incrementally on prior work.

---

> ### Author Rebuttal · Authors · 2026-03-30
>
> We sincerely thank the reviewer for the careful and generous reading. The three questions target real limitations and we address each with new data and honest assessment.
>
> ### Q1: Complex-valued PRISM Decoder
>
> We investigated this and concluded it is not practical with the current design. The core issue is that the Gated Harmonic Convolution performs circular convolution via FFT→H→IFFT. In this operation, every output position depends on all input positions simultaneously. There is no way to apply a causal mask after IFFT, because future information is already mixed into the result. There is also no way to enforce causality in the frequency domain, since every frequency bin is a global mixture of all time positions.
>
> In principle, one could convert circular convolution to causal linear convolution by zero-padding to 2L and constraining the filter to be one-sided in the time domain. However, our neural filter bypasses the convolution theorem entirely: it parameterizes H directly in the frequency domain via an MLP, rather than defining a time-domain kernel and FFT-ing it. There is no explicit time-domain kernel to constrain for causality. Enforcing that IFFT(H) is zero for negative indices would require a nontrivial spectral constraint on the MLP output. Even if this were solved for training (parallel, teacher-forced), autoregressive inference would require recomputing the full FFT for each new token, giving O(L² log L) total cost and removing the efficiency advantage entirely.
>
> We believe a proper complex-valued decoder would require either a causal reparameterization of the filter (an open research direction) or a recurrent formulation of the spectral convolution. We will include this discussion in the revision.
>
> ### Q2: Mixed Precision (BF16/FP32)
>
> We performed the requested benchmarks. The results clarify both the current practical cost and its root cause. Flash Attention is enabled for all models, and experiments were conducted on an A100 GPU with BF16 support. All benchmarks measure training throughput (forward, backward, and optimizer steps).
>
> **Table A: Realistic Deployment (BF16 autocast, batch=2)**
>
> | Model|Tok/s 4096|Tok/s 8192|Tok/s 16384|Mem GB 16384
> |---|---|---|---|---
> | Transformer | 237,248 | 215,703 | 166,419|16.6
> | FNet | 287,787 |300,555| 288,252 |16.1
> | PRISM | 118,793 |133,416|132,226|23.9
> | WPT | 95,341 |119,809 |103,376 |23.3
>
> **Table B: FP32 Parity (isolates architectural scaling)**
>
> |Model|Tok/s 4096|Tok/s 8192|Tok/s 16384| Mem GB 16384
> |---|---|---|---|---
> |Transformer|90,486|57,209|31,970|22.2|
> |FNet | 178,368 |148,736|105,378|22.1|
> |PRISM |91,602|87,861|70,857 |28.4|
> |WPT|63,956|47,677|30,013 |29.7|
>
> **Table C: Both confounds removed (FP32, Flash Attention off, Ours also OOMs at L=8192 without Flash Attention, as it contains attention in both the sensory stream and/or refiner.)**
>
> |Model|Tok/s 4096|Tok/s 8192
> |---|---|---
> |Transformer|70,796|OOM
> |FNet|161,212|115,581
> |PRISM| 86,946|72,954
> |WPT|56,341|OOM
>
> Note that under BF16, Transformers benefit from two framework advantages absent for PRISM: Flash Attention's fused CUDA kernels and native bf16 precision. Table B (FP32 parity) removes bf16 confound. Table C removes both confounds.
>
> Under BF16 autocast, PyTorch automatically keeps FFT/IFFT in FP32 (no complex BF16 kernel exists) while casting real-valued operations (gating, projections, refiner) to BF16. This is the best we can achieve within the current framework.
>
> The key finding is in Table B (and C). At FP32 parity, PRISM retains 77% of its throughput from L=4096 to L=16384 (91K→71K), while the Transformer retains only 35% (90K→32K). This confirms the O(N log N) vs O(N²) scaling advantage is real and architectural. The absolute throughput gap under BF16 (Table A) is a framework limitation: PyTorch lacks complex BF16 support entirely. This affects all complex-valued architectures equally and is not specific to PRISM.
>
> We will reframe the efficiency claim in the revision to separate architectural scaling (demonstrated) from absolute throughput (currently framework-limited).
>
> ### Q3: Mamba-3 Comparison
>
> We agree this would be valuable but could not complete a controlled comparison within the rebuttal period. Mamba-3 introduces confounding variables (recurrence, data-dependent gating) that make direct comparison nontrivial without ablating those components.
>
> ### Others
>
> We also found and fixed typographical errors you mentioned (e.g decoder typo instead of encoder) and completed ablations reviewers asked. We added explanations for specialized terms and softened the physics and neuroscience analogies while preserving the nature of our work.
>
> We agree with the reviewer that our ablations were limited, and we refer the reviewer to our discussion with reviewer xz3P for the corresponding results.
>
> We thank the reviewer again for this generous and constructive review.

---

> > ### Author Rebuttal · Reviewer_ykVX · 2026-04-03
> >
> > I acknowledge for authors' effort. My concerns have been addressed.

---

> > > ### Author Response · Authors · 2026-04-03
> > >
> > > Thank you for your thorough review, questions, and for engaging with our rebuttal. We deeply appreciate your time and your support of this work. We will ensure that the additional data, throughput tables, and clarifications discussed here are fully included in the final camera-ready version of the manuscript, if accepted.

---

### Official Review · Reviewer_zGvr · 2026-03-13

**Soundness:** 2
**Presentation:** 3
**Significance:** 2
**Originality:** 3
**Overall Recommendation:** 3
**Confidence:** 3

**Summary:**

This paper introduce a novel architecture that performs through phase rotation and inference rather than magnitude mixing in transformer based language models. To some extend this can be seen as spectral alternative to attention models similar to Fnet architecture but performing in complex domain. The authors provided intuitive and proper theoretical background for this idea however empirically the strongest result comes from a hybrid version of this architecture which is combined with standard architecture. While the motivation is strong but it makes it hard to dissect the exact mechanism is driving the gain for the reasoning in experiment hence validating the propsed theoretical idea on phase based inference.

**Compliance With Llm Reviewing Policy:**

Affirmed.

**Key Questions For Authors:**

1) the related work section point to relevant works like holographic transfromer and shows the however I'm curious if the PRISM provide any behavior above these existing models ?
2) It seems the authors restrict the FNet architecture from 2D to 1D and that also explains why the PRISM encoder parameter is reduced ( as a evidence for computational and spectral efficacy). Do the authors expect the performance of 1D Fnet is match the complicated architecture provided for PRISM?

**Limitations:**

1) the core empirical result is supported by hybrid architecture which complicate the key claim about suffciency of phase-only  computation provides a sufficient reasoning mechanism.

2) minor but important: the references can be enhanced for this work specifically due to the nature of the study design (which is meaningful combination of the existing earlier work on complex vector computation in transformers). Specifically this work will be relevant: complex vector attention: https://arxiv.org/abs/2505.10222

3) minor: the references can be enhanced for this work specifically due to the nature of the study design (which is meaningful combination of the existing earlier work on complex vector computation in transformers). Specifically this work will be relevant: complex vector attention: https://arxiv.org/abs/2505.10222

**Strengths And Weaknesses:**

Strengths:
1) the key idea is very intriguing. The authors clearly show the advantages of this method over holographic transformers.

2) The paper also introduce a coherent architecture desing with defined modules (phase interface layers, spectral updates, ..). This is valuable for concrete reproducibility. Also the hugging face for the implementations has been provided.

Weakness:
1) The paper repeatedly suggest subtractive reasoning is what is sufficient/or needed for reasoning but honestly the experimental result suggest something else: It's the hybrid approach that outperform other models.

2) intertwined ablations: the comparisons change multiple factors simultaneously (normalization, regularization, architecture variants). it is difficult to isolate whether improvements come from phase interference or other architectural differences.

---

> ### Author Rebuttal · Authors · 2026-03-30
>
> We thank the reviewer for the careful reading and helpful feedback. We agree that our original sufficiency claim was too strong. The hybrid model performs best, refining our interpretation. We will extend our related work section including the suggested Complex Vector Attention reference.
>
> ### 1. Reassessment of the central claim (Weakness 1)
>
> We agree the original claim was overstated. Phase interference is a parameter-efficient relational primitive, but language modeling requires both relational structure and token identity, leading to the hybrid design. Reviewer ykVX also noted this.
>
> In PRISM, the spectral branch is magnitude-invariant, so representations practically lie close to a unit circle. This suffices for some algebraic tasks but not language modeling, where magnitude encodes salience and confidence. Our L=1 collapse (Section 5.4) is consistent with this view. Phase also requires a dense spectrum for effective interference, explaining why a pure spectral model is insufficient for every case.
>
> **Revised claim:** Phase interference is a computationally efficient relational primitive (not reasoning) that, combined with thin attention, outperforms pure attention at lower cost at this scale. Our main contribution is investigating phase and wave mechanics in neural networks.
>
> ### 2. Causal isolation via filter ablation (Weakness 2)
>
> Following this suggestion (reviewer xz3P and ykVX also suggested this), we performed interventional experiments on the learned spectral filter H. Same checkpoint and weights, only H modified.
>
> | Condition | Constrained | Unconstrained (CReLU) |
> |---|---|---|
> | H = 1 (no filter) | 328× | 78× |
> | H = \|H\| (magnitude only) | 153× | 173× |
> | H = H / \|H\| (phase only) | **1.41×** | **1.51×** |
> | H = H* (phase reversed) | 245× | 139× |
> | Phase shuffled | 160× | 161× |
>
> Phase-only retains >95% of performance. Removing, reversing, or shuffling phase is catastrophic. This holds across two independently trained models (ModReLU+PPLN and CReLU+LN), confirming the behavior is not tied to specific parameterization.
>
> Notable: in the CReLU model, magnitude-only (173×) is worse than no filter (78×), indicating magnitude scaling depends on phase structure.
>
> We also tested CReLU-WPT (CReLU+LN in the relational stream): 5.46 PPL, worse than constrained WPT (4.94) despite CReLU being better standalone (5.80 vs 6.06). We hypothesize this is because CReLU can suppress noise via quadrant routing, reducing pressure to align phase angles precisely. In a hybrid where attention already handles token selection, ModReLU's constraint forces the phase stream to specialize in relational interference, which appears more complementary. This supports the relational primitive framing, though we present it as a hypothesis.
>
> ### 3. Real vs. complex architectural isolation
>
> A topology-matched real spectral encoder (identical structure, but using rfft/irfft with a 512→1024→512 expand-compress logic to match complex parameter capacity) achieves 6.15 PPL vs 5.80 for complex. The gap is attributable to persistent complex representations across layers.
>
> In WPT, replacing the PRISM stream with this real encoder (13.5M core params vs 12.9M): 5.55 PPL (real) vs 4.94 (WPT). A 12.3% gap from complex phase coding alone, despite the real variant having more core parameters. We refer reviewers to our response to Reviewer xz3P for the full ablation results.
>
> ### 4. FNet baseline (Question 2)
>
> Our FNet baseline uses 2D mixing, the stronger configuration. The real spectral encoder (6.15 PPL) vs FNet (9.87 PPL) shows improvement comes from our design, not Fourier transforms alone.
>
> ### 5. Holographic Transformers (Question 1)
>
> Holographic Transformers prescribe interference via explicit phase decay, coherent superposition, and regularization. PRISM provides only the spectral topology (FFT→H→IFFT) and interference emerges from training. Our filter ablation confirms this: structured phase rotations emerge even under phase-hostile conditions (CReLU+LayerNorm), suggesting wave-like computation is a natural attractor. Holographic Transformers also retain O(N²) attention, while PRISM uses O(N log N) filtering. We see these as complementary. We also would like to make this clear in our work if accepted.
>
> ### Summary
>
> The rebuttal ablations validate mechanisms already present in the submitted architecture. No architectural or mathematical changes were needed. The filter interventions isolate a single variable (H), confirming phase drives performance. The revised claim refines rather than changes the contribution. Revisions (softened claims, added ablations, diagrams, code) fall within normal scope.

---

> > ### Author Rebuttal · Reviewer_zGvr · 2026-04-04
> >
> > Thanks to the authors for the detailed rebuttal. I believe these changes substantially strengthens the paper, particularly through the interventional filter ablation. This directly addresses my earlier concern regarding causal isolation and is a strong addition. Also authors clarification and reframing phase interference as a relational primitive rather than a sufficient mechanism for reasoning is appreciated. This is a more precise and defensible interpretation of the results.
> >
> > I collectively increase the soundness score to 3.

---

> > > ### Author Response · Authors · 2026-04-04
> > >
> > > Thank you for your engagement. You selected "partially resolved" and mentioned follow-up questions, but both weaknesses in your original review (causal isolation and intertwined ablations) appear addressed based on your own acknowledgment. Could you clarify what remains unresolved?
> > >
> > > We also encourage you to review the additional causal ablations in our responses to Reviewer xz3P (layer-wise phase/magnitude scrambling, real vs. complex matched comparisons, WPT component ablation), which we believe are directly relevant to your concerns
> > >
> > > Finally, the soundness score on the form still reads 2, though your text states an increase to 3. We wanted to flag this in case it was an oversight.
> > >
> > > Best regards,
> > >
> > > The Authors

---

### Official Review · Reviewer_wyB3 · 2026-03-15

**Soundness:** 3
**Presentation:** 2
**Significance:** 3
**Originality:** 3
**Overall Recommendation:** 4
**Confidence:** 2

**Summary:**

This paper proposes a new complex valued architecture that tries to removes (at least partially) the dependency on the magnitude and focus on the phase of the complex value. Intriguing, this constraint is physically grounded. The authors further train this architecture and some hybrid architecture with this architecture and transformers on language modeling and found one of the hybrid architecture achieves sample efficiency improvement over transformer architecture.

**Compliance With Llm Reviewing Policy:**

Affirmed.

**Final Justification:**

The rebuttal addresses my main concerns on the hybrid architecture ablation

**Key Questions For Authors:**

**Q1.** In Fig 2, both antonyms and synonyms show strong phase locking, will antonyms to show negative instead of positive correlations if absolute value is not taken in equation 13.

**Q2.** Regarding the hybrid architecture, how does the interpretability metrics in Section 5 when additional attention layers are introduced?

**Limitations:**

yes

**Strengths And Weaknesses:**

**Soundness:** The motivation to reduce the dependency on magnitude to focus more on phase for complex valued network makes sense.  The paper design the architecture to softly controls the magnitude and is very clear about the architecture design. In the experiment, multiple different architectures are considered and there are some ambiguity about the choices of hybrid architectures. Especially, it is unclear why WPT can beat the baseline while HSSM is worse than the PRISM architecture.


 **Presentation:** The visualization in Figure 1 is informative about the pure PRISM architecture. Meanwhile, the paper can be improved by incorporating visualization about the hybrid architecture.


 **Significance & Originality:** Without too much prior knowledge on complex-valued network, the reviewer thinks this is a novel attempt in trying to study the importance of phase in complex value network.

---

> ### Author Rebuttal · Authors · 2026-03-30
>
> We thank the reviewer for the thoughtful review and creative questions. Your Q1 directly inspired a new analysis that strengthens our understanding of the Phase Compass.
>
> **Q1 (Antonyms without absolute value).** We checked this. A creative intuition, but the data shows antonyms still exhibit positive coherence (mean R = 0.125, CI [0.104, 0.149]), not anti-phase. We ran BERT as a sanity check and observed similar behavior. This is expected from distributional semantics: "cold" and "hot" appear in nearly identical contexts ("the tea is hot/cold"), so embeddings encode them in shared spectral neighborhoods. The Phase Compass acts as a topic binder, grouping semantically related concepts, rather than a polarity classifier. This motivates our shift toward causal, interventional evidence (see below).
>
> **Q2 (Interpretability in hybrids).** The hybrid architectures use Dynamic RoSE, which introduces content-dependent phase steering. This makes the static pairwise R metric from Section 5 difficult to apply directly. We can train a hybrid on WMT14 with the validated pipeline upon request.
>
> However, our mechanistic evidence now goes well beyond embedding analysis. In response to ablation requests from Reviewers xz3P and zGvr, we performed causal interventions directly on the learned spectral filters H in the FFT→H→IFFT pipeline. Same checkpoint, same weights, one variable changed:
>
> | Condition | Constrained (ModReLU) | Unconstrained (CReLU) |
> |---|---|---|
> | H = 1 (no filter) | 328× | 78× |
> | H = |H| (magnitude only) | 153× | 173× |
> | H = H/|H| (phase only) | **1.41×** | **1.51×** |
> | H = H* (phase reversed) | 245× | 139× |
> | Phase shuffled | 160× | 161× |
>
> Phase-only filters preserve >95% of performance. Removing, reversing, or shuffling phase is catastrophic. This holds identically across two independently trained models with different activations and normalization, confirming the mechanism arises from the spectral topology, not from our specific architectural constraints.
>
> Crucially, we discovered that our phase-preserving components (ModReLU, PPLN) are not what forces the model to use phase but model prefers using phase despite our phase-destroying ablations. We replaced all phase-preserving components with phase-destroying alternatives (CReLU, standard LayerNorm), and performance actually improved (5.80 vs 6.06 PPL). A topology-matched real-valued spectral encoder achieves 6.15 PPL. The gap from 6.15 to 5.80 is strictly attributable to persistent complex representations across layers, and the filter ablation confirms these representations are used for frequency-domain interference.
>
> **On HSSM vs WPT.** The pattern across our results is consistent: Transformer > PRISM > FNet as standalone encoders. Parallel Hybridization improves both directions. HSSM pairs PRISM with FNet, and WPT pairs PRISM with Transformer. The difference is that HSSM's passive FNet stream collapses at L=4096 (9.87 PPL standalone), limiting the hybrid. WPT's active attention stream can dynamically select content, complementing the phase stream effectively.
>
> Additionally, several reviewers (zGvr, ykVX, xz3P) noted that our ablations were limited. We have completed the requested ablations, and the results now clearly demonstrate the contribution of the phase component. Detailed ablation results are provided in our discussion with reviewers zGvr and xz3P.
>
> To isolate whether the gain comes from phase specifically, we replaced WPT's PRISM stream with a parameter-matched real spectral encoder (13.5M core params vs 12.9M). Result: 5.55 PPL (real) vs 4.94 PPL (WPT). A 12.3% gap from complex phase coding in the relational stream alone.
>
> **Presentation.** We will include hybrid architecture diagrams and clarify the relationship between HSSM, WPT, and the standalone models in the revised version, as suggested. We also softened physics analogies and overclaims expressed by reviewer zGvr and xz3P.

---

> > ### Author Rebuttal · Reviewer_wyB3 · 2026-04-05
> >
> > The authors' rebuttal addresses most of my concerns and I will include my score to 4.

---

> > > ### Author Response · Authors · 2026-04-05
> > >
> > > Thank you for taking the time to review our rebuttal and for updating your score. We’re really glad that the additional analyses helped clarify things and address your concerns. We will include these new analyses in the paper if accepted.
> > >
> > > Best Regards,
> > >
> > > The Authors

---

### Decision · Program_Chairs · 2026-04-30

**Decision:**

Accept (regular)

**Comment:**

Reviewers agree this paper presents a novel contribution to the study of the computational aspects of language modeling and that the proposed architecture is an interesting way to take advantage of these findings. During rebuttals the authors ran extensive further experiments and it appears all reviewers concerns have been adequately addressed.